# Autophagy Regulation on Cancer Stem Cell Maintenance, Metastasis, and Therapy Resistance

**DOI:** 10.3390/cancers14020381

**Published:** 2022-01-13

**Authors:** Xin Wang, Jihye Lee, Changqing Xie

**Affiliations:** Thoracic and GI Malignancies Branch, Center for Cancer Research, National Cancer Institute, National Institutes of Health, Bethesda, MD 20814, USA; wangx49@mail.nih.gov (X.W.); leej64@mail.nih.gov (J.L.)

**Keywords:** autophagy, cancer stem cells, stemness, self-renewal, metastasis, treatment resistance

## Abstract

**Simple Summary:**

Certain types of cancer have higher relapse rates compared to others, and cancer stem cells (CSCs) have been shown as the main drivers of cancer relapse and cancer severity. This subpopulation of cells displays stem-like characteristics which bolster tumorigenesis along with metastasis and lead to poorer prognoses. Autophagy has been studied as a mechanism by which CSCs maintain stemness and acquire resistance to chemotherapy and radiation. The aim of this review is to condense and organize what has been recently published on the connection between cancer stem cells (CSCs) and autophagy. Multiple studies on autophagy have suggested that the pathway is a double-edged sword, which can either undermine or enhance CSC characteristics depending on interactions with different pathways. Thus, future research should investigate regulation of autophagy in combination with traditional cancer therapies as a possible method to effectively eliminate CSCs and minimize cancer relapse.

**Abstract:**

Cancer stem cells (CSCs) are a subset of the tumor population that play critical roles in tumorigenicity, metastasis, and relapse. A key feature of CSCs is their resistance to numerous therapeutic strategies which include chemotherapy, radiation, and immune checkpoint inhibitors. In recent years, there is a growing body of literature that suggests a link between CSC maintenance and autophagy, a mechanism to recycle intracellular components during moments of environmental stress, especially since CSCs thrive in a tumor microenvironment that is plagued with hypoxia, acidosis, and lack of nutrients. Autophagy activation has been shown to aid in the upkeep of a stemness state along with bolstering resistance to cancer treatment. However, recent studies have also suggested that autophagy is a double-edged sword with anti-tumorigenic properties under certain circumstances. This review summarizes and integrates what has been published in the literature in terms of what role autophagy plays in stemness maintenance of CSCs and suggests that there is a more complex interplay between autophagy and apoptosis which involves multiple pathways of regulation. Future cancer therapy strategies are needed to eradicate this resistant subset of the cell population through autophagy regulation.

## 1. Cancer Stem Cells

CSCs, also known as tumor initiating cells (TIC) and cancer initiating cells (CIC), are a select subset within the tumor population that have been demonstrated to initiate tumor formation, tumorigenicity, resistance to different types of treatment, and metastasis [1]. CSCs have also been insinuated as the primary culprit for cancer severity and cancer relapse in various solid tumors, including breast, ovarian, brain, lung, pancreatic, liver, and colorectal cancers [2,3,4,5,6]. CSCs are capable of remodeling the tumor microenvironment (TME) through the use of matrix metalloproteinases (MMP), which are calcium-dependent zinc-containing endopeptidases induced by hypoxia and extracellular acidosis [1,3,7,8,9]. TME reconstruction enhances cancer invasion and metastasis, allowing CSCs to undergo epithelial-mesenchymal transitions (EMT), where the cancer cells lose epithelial characteristics and develop mesenchymal features that allow for invasion into the local vascular network and migration to distant locations with more optimal conditions for tumor growth [10]. Further, CSCs are able to escape immune surveillance due to their reduced expression of tumor antigens, impaired antigen presentation machinery, and ability to recruit immunosuppressive factors via secreted inflammatory molecules [10,11,12].

The origin of CSCs has not been completely elaborated upon; however, there are several theories on this topic. CSCs are shown to stem from normal stem cells that acquire CSC characteristics triggered by environmental changes (i.e., chronic inflammation stimulated through enriched secretion of chemokines, cytokines, and growth factors) which induce genetic mutations [9]. Moreover, environmental regulators, such as ageing, obesity, and a high fat diet, have been shown to correlate with a heightened inflammation state. This can induce nonstem-like circulating cancer cells to de-differentiate into CSCs [1,13]. Cancer cells may also acquire malignant, stem-like characteristics over many sequential divisions. Additionally, dysregulated epigenetic changes, including hypermethylation of multiple CpG islands and downregulation of DNA methylation enzymes, can cause normal cells to reprogram back to a more un-differentiated stem-like state [1]. Together, these alterations illustrate the various mechanisms that allow for the development of CSCs.

These cells have characteristic stem cell properties such as indefinite self-renewal and differentiation potential to become mature terminal malignant cells [1,9]. CSCs are also known to have cellular plasticity, which is the ability to take on different phenotypic characteristics via processes such as dedifferentiation. This process can be seen in normal differentiated cells under chronic stress [14]. However, CSCs take advantage of this ability to differentiate into various cell types that make up the tumor. Additionally, already differentiated cells may de-differentiate into different tumor cells, while a small subsection of CSCs remains in a stem-like state. Thus, the dynamic nature of CSC plasticity creates what is termed heterogeneity, which is how tumors are composed of various types of cells that are further distinguished from each other by different genetic mutations [1]. A prime example is epithelial-mesenchymal plasticity, where cells upregulate EMT transcription factors Snail, Twist, Zeb, and vimentin, while downregulating E-cadherin. This induces the cells to lose their epithelial phenotype to take on a more mobile mesenchymal morphology [14,15].

There is no consensus on which transcription factors regulate self-renewal across different CSC lines. Embryonic stem cell transcription factors, such as Nanog and sex determining region Y-Box Transcription Factor 2 (SOX2), have been shown to help maintain hepatocellular carcinoma (HCC) and oligodendroglioma CSC self-renewal [16,17,18].

Alterations in multiple signaling pathways have been found to regulate CSCs properties. A study demonstrated colorectal CSCs with enhanced activation of Wnt/β-catenin, which is a signaling pathway known to regulate various cell processes including cell proliferation, differentiation, apoptosis, and tissue homeostasis [19]. Aberrant Wnt/β-catenin signaling subsequently enhanced CSC properties such as expression of cell surface markers, self-renewal, and tumorigenicity [20]. The Notch pathway plays a critical role in regulating cell differentiation, apoptosis, and migration during the normal biological development of an organism along with maintenance of homeostasis [21]. Aberrant activation of Notch signaling has been shown to promote self-renewal in breast CSCs [22,23]. The conserved Sonic Hedgehog signaling pathway modulates a critical stage in the development of an organism, which includes embryonic development, tissue repair, and cell differentiation (this specifically pertaining to gastrointestinal cell lineages) [24,25]. Gene expression analysis of bladder CSCs revealed that upregulation of the Sonic Hedgehog pathway was crucial for self-renewal and population expansion [26]. Aberrant Hedgehog signaling was also found to enhance self-renewal and resistance to apoptosis in lung adenocarcinoma CSCs [27]. Furthermore, the NF-κB family of transcription factors regulates expression of multiple genes in response to various diseases. These responses include induction of inflammation, cell proliferation, angiogenesis, and metastasis [28,29]. In breast CSCs, the NF-κB signaling cascade has been demonstrated to enhance self-renewal and tumorigenicity [30]. Furthermore, the JAK2/STAT3 signaling pathway is involved with many processes such as cell proliferation, cell differentiation, and modulation of the immune response to infection [31]. Cytokine-receptor binding activates JAK to phosphorylate specific STATs to alter the expression of certain genes that regulate biological processes [32]. JAK2/STAT3 has been shown to regulate self-renewal and tumorigenicity in glioma CSCs [33]. TGFβ/Smad signaling is another pathway that regulates multiple biological processes such as cell-cycle arrest, apoptosis, cell proliferation, cell differentiation, and immune inhibition [34,35]. TGFβ/Smad has been shown to regulate stemness, self-renewal, and chemosensitivity in liver CSCs [36]. Lastly, the Hippo/YAP signaling pathway has been found to be associated with regulation of tissue growth, the immune system, tissue development during early mammalian development, and tissue regeneration/wound healing [37]. Hippo/YAP signaling has been demonstrated to promote self-renewal maintenance in liver and esophageal CSCs [38,39] and drug resistance in urothelial CSCs [40]. Together, these studies suggest that there are multiple signaling pathways that can be activated and may even work synchronously to regulate CSC properties such as self-renewal.

There are several methods to identity CSCs which include CSC markers (e.g., CD24, CD44, CD90, CD133), high aldehyde dehydrogenase (ALDH) expression, spheroid formation, and in-vivo tumor formation with a small seed number. The latter is considered the gold standard for determining tumorigenic ability [1,16,41].

Compared to normal cells, CSCs have an altered metabolism which was coined the Warburg effect by Otto Warburg [42,43]. CSCs have reduced mitochondrial oxidative phosphorylation, while upregulating glycolysis. This results in the accumulation of extracellular lactate, resulting in extracellular acidosis, which is a lower TME pH due to the agglomeration of H+ ions. Furthermore, CSCs upregulate glutathione synthesis to guard against reactive oxygen species (ROS) and escape cell death [42]. Another barrier that coincides with tumor growth is that CSCs are theorized to reside in the tumor core, which is plagued with low pO2 levels (hypoxia) and paucity of nutrients [44,45]. In recent years, autophagy has come into the spotlight as a possible mechanism to recycle nutrients, feed the growing tumor, and regulate maintenance of stemness in CSCs.

## 2. Autophagy

Autophagy is a mechanism of energy metabolism where intracellular components including damaged organelles, pathogens, and non-essential proteins are sequestered into vesicles and recycled during moments of environmental stress such as nutrient deprivation, oxidative stress, hypoxia, and infection in order to provide nutrients and overcome new stressors [8,46]. Autophagy can be broken down into three main groups: macro-autophagy, micro-autophagy, and chaperone-mediated autophagy. Macro-autophagy is characterized by the envelopment of materials by endoplasmic reticulum-derived membranes followed by fusion with lysosomes for degradation [47]. Micro-autophagy is the direct engulfment of cytoplasmic material, which is mediated by the lysosomes [48]. Lastly, chaperone-mediated autophagy is when chaperone proteins bind and transport intracellular proteins to the lysosomes [49]. This paper will mainly focus on macro-autophagy and will subsequently use the convention of referring to this process as “autophagy.” Interestingly, there are two different roles for macro-autophagy reported in the literature. Lethal/toxic autophagy, also known as autophagic cell death, results in increased cell death, which has been reported in glioblastoma and HCC [50,51]. Conversely, protective autophagy is a major cause of survival and chemotherapy resistance in CSC populations [15]. This form of autophagy promotes metastasis, allowing the cancer to spread and wreak havoc throughout the body. This was demonstrated in pancreatic, colorectal, and lung cancers [5,6,52].

Initiation of autophagy begins with the formation of a non-degradative vesicle called the autophagosome. This vesicle then fuses with a lysosome to form the autolysosome, where intracellular materials are degraded and released into the cytoplasm for recycling [4,46]. Autophagosome generation occurs via a three-step process; initiation starts with autophagic machinery assembling at the membrane source site to form the pre-autophagosome membrane. Two main multi-protein complexes, Unc-51-like autophagy activating kinase (ULK) complex and vacuolar protein sorting 34-beclin 1 (VSP34-BECN1) complex, are both needed for initiation and nucleation of the isolation membrane/phagophore, which is a double membrane that encloses cytosolic material [46,53]. Autophagy related protein 9 (ATG9) recruits lipids for the isolation membrane [54]. Protease ATG4 catalyzes the cleavage of microtubule-associated protein light chain 3 (LC3) to LC3-I. ATG7, ATG3, and the ATG5-ATG12-ATG16L1 complex covalently conjugate phosphatidyl-ethanolamine (PE) with LC3-I, forming LC3-II [2]. LC3-II relocates to the nucleation membrane to initiate elongation and closure of the autophagosome around intracellular materials. Finally, SNARE-like proteins promote the fusion of lysosomes with autophagosomes to form autolysosomes (Figure 1) [46].

The autophagy pathway has many points of regulation. It is mainly suppressed by mammalian target of rapamycin (mTOR), which inhibits ULK complex activation [15,55]. Additionally, Bcl-2 family proteins inhibit VSP34-BECN1 complex activation [56]. A recent paper demonstrated in a colorectal carcinoma (CRC) model that the YAP/TEAD complex binds to upregulate Bcl-2 expression, which leads to autophagy inhibition [57]. This suggests that additional pathways may play a role in autophagy regulation.

The role of autophagy in cancer has been proposed as a double-edged sword which inhibits early-stage tumorigenesis. However, in the later stages, autophagy has been demonstrated to enhance tumor preservation, growth, and proliferation in a variety of tumor types, including prostate, breast, and ovarian cancers [58,59,60]. Studies on autophagy have developed various targets on different sections of autophagosome formation. For example, beclin1 KD or 3-MA (autophagy inhibitor) treatment inhibits early autophagy activation, while ATG7, ATG5, or ATG12 KD prevents elongation and sealing of the autophagosome. Drugs such as BafA1 and chloroquine (CQ) inhibit later stages of autophagy by preventing fusion of autophagosomes with lysosomes [61]. In fact, many clinical trials are looking at regulation of autophagy as a means to treat cancer (Table 1).

## 3. Pro-Survival Autophagy Promotes Stemness Maintenance

In recent years, research has looked at the complex role that autophagy plays in CSC function. Most of the literature suggests that CSCs utilize autophagy as a pro-survival mechanism to maintain a dormant-like stage, which allows for resistance against environmental stresses including hypoxia, chemotherapy, and radiotherapy [2,4,46].

A study on teratocarcinoma cells (cancer of the embryonic stem cells) investigated how alterations in autophagy levels affect CSCs. Inhibition of Nicotinamide phosphoribosyl transferase (NAMPT) activated PTEN, an inhibitor of mTOR, and subsequently induced autophagy activation, evidenced by elevated mRNA and protein levels of autophagy-related genes including ATG5, ATG7, and LC3-II. Subsequent alterations included reduced POU domain transcription factor (POU5F1) expression along with other pluripotency factors such as Nanog and SOX2, resulting in lower proliferation and augmented differentiation of CSCs. Inhibition of autophagy by knocking down ATG12 and ATG7 also reduced stemness and promoted differentiation and/or senescence, suggesting that autophagy supports CSC stemness and blocking autophagy may be a potential target to eliminate teratocarcinoma stem CSCs [62].

Pagotto et al., (2017) explored inhibition of autophagy via ATG5 KO or CQ treatment in CD118+CD44+ ovarian CSCs. The treatment not only impaired cell viability but also reduced spheroid formation, downregulated stem cell-associated markers (e.g., Nanog, SOX2, OCT4), and tumorigenic potential, suggesting that autophagy enhances ovarian CSC stemness [2]. The findings of Li et al., (2017) showed that autophagy inhibition via ATG3 KD, ATG7 KD, or CQ significantly reduced the proportion of hepatic Axin2+CD90+ CSCs. Autophagy inhibition was also found to reduce expression of hepatocyte growth factor (HGF). HGF activates the receptor c-Met to induce JNK/STAT3 signaling, which upregulates CSC self-renewal and tumorigenesis, further corroborating the fact that autophagy plays some role in CSC stemness maintenance [63,64].

The mechanism by which autophagy promotes CSC self-renewal is still under investigation; however, a recent study suggests that p53 may play an important role in this regard. It is well-known that changes in p53 intracellular localization affect autophagy regulation. P53 nuclear localization allows the transcription factor to upregulate expression of autophagy-related genes, while undermining negative regulators of autophagy (i.e., P13K, AKT, and mTOR) [65]. However, p53 cytoplasmic localization inhibits the activity of AMP-dependent kinase (AMPK). Thus, this allows for mTOR activation and prevents autophagy activation [66]. Recently, Wang et al., (2021) showed that autophagy post-transcriptionally regulates p53 levels by removing cytosolic ub-p53 in the lung CSC cell line A549 and augments CSC stemness and spheroid/tumor formation [61].

Mitophagy is a subset of autophagy based on DPR1-driven targeted removal of dysfunctional mitochondria, often marked by increased oxidative stress, DNA damage, and accumulation of p62 protein [67]. Mitophagy works to maintain low ROS levels, which prevents excessive ROS-induced genome damage and consequent induction of mitochondria-dependent apoptosis [68,69]. Cell plasticity has been shown to be accompanied with significant changes in mitochondrial composition, function, and maturation, suggesting that mitophagy is crucial for maintenance of stemness [69]. Pten-induced putative kinase 1 (PINK1) has been shown to maintain mitochondrial morphology and function along with mitophagy-dependent mitochondrial degradation [70]. Vazquez-Martin et al., (2016) showed that KD of PINK1 inhibited mitochondrial rejuvenation and prompted spontaneous differentiation of induced pluripotent stem cells (iPSCs). Thus, this suggests that PINK1-dependent mitophagy is required for iPSC to maintain an undifferentiated state [69]. Liu et al., (2017) illustrated that mitophagy regulates tumor suppressor p53 levels in hepatic CSCs. Blockade of autophagy (ATG5 KD, ATG7 KD, and 3-MA) results in increased PINK1-dependent phosphorylation of p53 at serine 392, which induces stabilization and tetramer formation. Phosphorylated p53 is able to translocate to the nucleus to bind to the Nanog promoter and act as a competitive inhibitor against OCT4-SOX2 attachment. Therefore, this suggests that mitophagy removes mitochondria-associated and phosphorylated p53 to allow for Nanog expression and enhances hepatic CSC self-renewal and stemness maintenance [71]. Together, these studies suggest that pro-survival autophagy can help enhance and maintain stemness in various CSC types (Table 2).

## 4. Pro-Survival Hypoxia-Induced Autophagy Promotes Metastasis

CSCs constantly modify their TME to create a more suitable niche for tumor growth. One common alteration is the creation of hypoxic areas. The high proliferation rate of tumor cells places great stress on the local vasculature and strips oxygen away from the environment. Thus, hypoxia often induces autophagy as a means to recycle nutrients. One study on CD133+ pancreatic CSCs demonstrated that a hypoxic environment induced greater spheroid formation and enhanced expression of OCT4 and SOX2, which indicates greater self-renewal capacity. Upregulated levels of hypoxia-inducible factor 1a (HIF-1α), the main transcription factor for hypoxia-induced genes [77], resulted in activation of autophagy, which was measured by increased expression of beclin1, ATG5, and LC3-II. In addition, HIF-1α siRNA along with 3-MA treatment resulted in mesenchymal-epithelial transition (MET), where the cells lost their mobile fibroblast phenotype and took on epithelial characteristics. These cells were observed to have increased expression of E-cadherin with downregulation of mesenchymal markers, vimentin and MMP-9. Thus, this study suggests that hypoxia-induced autophagy may play a role in both CSC self-renewal and metastasis [5] (Table 2).

## 5. Pro-Survival Autophagy Promotes Treatment Resistance

CSCs have been documented to resist conventional cancer therapies, such as chemotherapy and radiation, and subsequently are able to reestablish tumors after treatment when CSCs are motivated by minute stimuli in the TME that trigger signaling cascades [78]. In recent years, autophagy has been reported as a mechanism that grants resistance to cancer treatment in CSCs.

Radiotherapy (RT) targets tumor cells with photons (e.g., X-ray) and particles (e.g., electrons, protons, and heavy ions) to directly induce irreparable DNA damage by DNA ionization or indirectly through the generation of ROS by interacting with water molecules [72,73]. CSCs have been shown to elevate levels of the antioxidant glutathione (GSH) to protect against ROS-induced oxidative stress [79]. The CSC marker SLC3A2/CD98 heavy chain (CD98hc) associates with cysteine transporter SCL7A11 to synthesize GSH. Digomann et al., (2019) found that amino acid restriction in CD98hc KO HNSCC CSCs induced inhibition of mTOR/PI3K and subsequently activated autophagy. Autophagy inhibition via ATG5 KD or BafA1 increased radio sensitization and induction of apoptosis [72]. Another study on breast CSCs demonstrated increased autophagic vesicles paired with a delayed response to irradiation and reduced initial apoptosis induction compared to normal cancer cells [73]. Both studies suggest that autophagy may play a protective role against radiotherapy for CSCs.

In-vivo murine models have demonstrated that cotreatment with Irinotecan (chemotherapy) and autophagy inhibition, CQ, significantly reduced tumor size compared to any one therapy alone [6]. Irinotecan is hydrolyzed into SN-38, a topoisomerase I inhibitor. By preventing the function of topoisomerase I, this leads to inhibition of DNA replication and transcription, thus preventing tumor cells from replicating [80]. A recent study showed that Taxol (chemotherapy) treatment induced activation of the ERK pathway and increased expression of autophagy-related proteins in radio-resistant bladder CSCs but not in regular cancer cells [74]. Taxol works by stabilizing microtubules and prevents progression of mitosis. In addition, this hinderance induces the cell to undergo apoptosis or transition to the G0 phase [81]. Taxol paired with autophagy inhibition by beclin1 and ATG5 KD resulted in increased apoptosis [74]. Interestingly, Brunel et al., (2021) noted that Temozolomide (chemotherapy) treatment reduced proliferation of beclin1 KD cells glioblastoma stem cells (GSCs) without enhancing apoptosis, which suggests that there may be other pathways besides autophagy that promote resistance to chemotherapy [4]. Temozolomide works as a DNA methylguanine-DNA methyltransferase (MGMT); however, in many tumor cells, these genes are epigenetically silenced. Thus, the hypermethylation cannot be removed, which induces DNA damage and triggers apoptosis [82,83]. Zhu et al., (2021) elucidated that CSCs can resist chemotherapy through SOX2-dependent upregulation of ATP-binding cassette (ABC) transporters, which serve as chemotherapy efflux pumps [84,85]. SOX2 synergistically co-localizes with β-catenin to enhance ABC transporter promoter activity but also binds to the beclin1 promoter to induce autophagosome formation. Thus, CSCs are able to effectively pump out chemotherapy drugs before the drugs are able to take effect. Further, autophagy activation by rapamycin augmented chemoresistance, stemness, proliferation, and invasion, while autophagy inhibition with 3-MA diminished the malignant cancer phenotype [6]. Thus, both studies corroborating that autophagy strengthens CSC resistance to chemotherapy.

Targeted treatments are another subset of cancer treatment. A recent paper suggests that in HNSCC CSCs, the tyrosine kinase inhibitor afatinib increased ROS levels, which activated REDD1-TSC1 signaling to inhibit mTOR complex 1 (mTORC1) and subsequently induces autophagy activation. Afatinib coincubation with autophagy inhibitors, 3-MA or CQ, illustrated augmented levels of afatinib-induced apoptosis, suggesting that afatinib can induce pro-survival autophagy in CSCs by reducing sensitivity to apoptosis.This validates the need for further research into autophagy inhibition and conjoint therapies to more effectively eradicate CSCs [75]. Overall, these studies have demonstrated that autophagy enhances resistance toward conventional cancer treatment modalities (Table 2).

## 6. Lethal Autophagy and Cancer Stem Cells

Some studies have indicated that autophagy is involved with suppression of cancer progression as demonstrated in gastric cancers and NSCLC [86,87]. Recent studies have also denoted that autophagy may negatively impact CSCs and cancer growth, hence termed lethal autophagy [15]. Curcumin, an anti-inflammatory spice, exhibited induction of lethal autophagy, followed by reduced glioblastoma SC (GSC) self-renewal and induction of differentiation in mice [76]. Tao et al., (2018) noted that mTOR inhibition (e.g., AZD8055 and rapamycin) suppressed self-renewal ability and tumorigenicity in CD133+Nestin+ GSCs, while 3-MA treatment reversed the mTOR inhibition phenotypes. Additionally, in-vivo mTOR suppression treatment in mice demonstrated reduced tumor size and prolonged survival through degradation of Notch1, which is involved in cancer cell tumorigenicity, self-renewal, and other various biological functions [50]. Similarly, autophagy inhibition via beclin1 KD, ATG5 KD, or CQ treatment on GSCs enhanced expression of stemness markers (e.g., CD133, POU5F1, SOX2, BMI1, LGR5, and NANOG), along with increased proliferation and clonogenicity. This suggests a departure from a dormant stemness status to an active, proliferating state when autophagy is inhibited [4,50].

Another study reinforces that autophagy inhibits dedifferentiation in a hepatocyte model. Ductular reaction is liver regeneration following chronic liver injury where the hepatocyte regeneration capacity has been impaired. Hepatocytes are able to dedifferentiate into ductular cells, which are liver progenitor cells (LPCs) [88]. Autophagy inhibition (ATG5/ATG7 KO) upregulated expression of YAP and Transcriptional coactivator PDZ-binding motif (TAZ), which are essential for controlling organ size and stemness in hepatocytes. Autophagy-deficient livers illustrated that strong YAP and TAZ co-expression induces ductular cell formation and carcinogenesis of HCC. Additionally, autophagy-deficient livers had significant accumulation of cleaved caspase-3 and ductular markers, including SOX9, cytokeratin 19, panCK, suggesting that autophagy prevents apoptosis in hepatocytes and undermines dedifferentiation of hepatocytes into LPCs [51] (Table 2). Together, these studies support the idea that autophagy impedes CSC stemness and proliferation (Figure 2).

## 7. Conclusions and Future Directions

Clearly, there is a discrepancy in the literature over the role of autophagy in CSCs. This review mainly focused on how autophagy strengthens CSC self-renewal, maintenance of a pluripotent state, and resistance against cancer treatment. However, there is a growing body of literature that argues that autophagy has the opposite effect and may inhibit tumorigenesis. We hypothesize that the type of autophagy differs based on the cell type, tumor stage, TME, and interaction with the apoptosis pathway. Multiple pathways are known to regulate autophagy and apoptosis; e.g., PI3K/AKT/mTOR axis inhibits autophagy but also phosphorylates caspase-3, caspase-9, and Bad to hinder apoptosis. Furthermore, autophagy and apoptosis can work synergistically to induce programmed cell death or antagonistically to allow for cell survival [89]. Therefore, the effect of autophagy on CSC characteristics may depend on which pathways interact with each other.

Given the amount of research on the interaction between autophagy and CSC/cancer progression, there is a subset of ongoing clinical trials focused on regulating autophagy in combination with other cancer treatments. Once again, there is a divide in the treatments where some studies focus on autophagy inhibition with CQ/hydroxychloroquine, while others inhibit mTOR with rapamycin, which subsequently allows for activation of autophagy. Only one clinical trial specifically focuses on CSCs, while the majority of trials study drug efficacy in cancer in general (Table 1). Thus, there is an urgent need to further elucidate the role of autophagy in CSCs. Interestingly, many regulatory factors of autophagy, e.g., beclin1, p62, and Bcl-2 family proteins, also play a role in apoptosis regulation.

Autophagy genes have also been observed to have non-autophagy related functions such as LC3-associated phagocytosis (LAP), which is when LC3 is conjugated to phagosome membranes. LAP activation is independent of nutrient deprivation, whereas canonical autophagy requires starvation. LAP has been shown to regulate inflammation and the immune response in response to fungal infection. Moreover, beclin1 has been demonstrated to have non-autophagy functions such as regulation of endocytic receptor trafficking, apoptosis, and inflammation [90]. One study illustrated that beclin1 is able to regulate growth factor and nutrient receptors, thus inhibiting tumor proliferation [91]. Therefore, it is crucial to clarify the role that autophagy plays in CSCs and cancer development.

Instead of knocking out entire autophagy genes, future studies are needed to focus on creating specific autophagy-deficient mutant alleles where non-autophagy functions are preserved, while autophagy is inhibited. This process will clarify the true role autophagy plays in CSCs. Additionally, the interaction between autophagy and apoptosis and how this affects CSCs should be investigated. Together, these studies should elucidate whether autophagy regulation paired with traditional cancer therapies are beneficial for patient outcome. Overall, unraveling autophagy’s role in CSCs may offer novel treatments and antineoplastic combination therapies to effectively eradicate the CSC population and improve the success rate of adjuvant therapies.

## Figures and Tables

**Figure 1 cancers-14-00381-f001:**
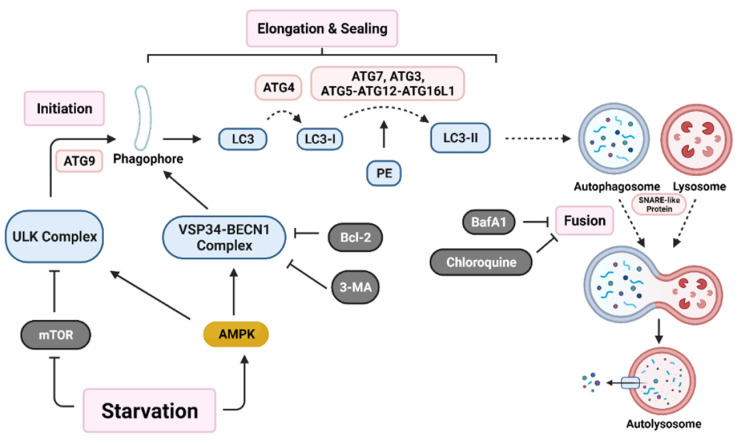
Autophagy activation pathway. Nutrient deprivation induces nucleation of the phagophore through activation of both the Unc-51 like autophagy activating kinase (ULK) complex and the vacuolar protein sorting 34-beclin 1 (VSP34-BECN1) complex. Various autophagy related gene (ATG) proteins work together to elongate and seal the phagophore through cleavage of microtubule-associated protein light chain 3 (LC3) into LC3-I and conjugation with phosphatidyl-ethanolamine (PE) to form LC3-II. Lastly, SNARE-like proteins aid in the fusion of autophagosomes with lysosomes resulting in an autolysosome. Various proteins (B-cell lymphoma 2 (Bcl-2)) and drugs (3-methyladenine (3-MA), bafilomycin A1 (BafA1), and chloroquine) are able to inhibit early-stage or late-stage autophagy induction. (Created with BioRender.com, accessed on 11 October 2021).

**Figure 2 cancers-14-00381-f002:**
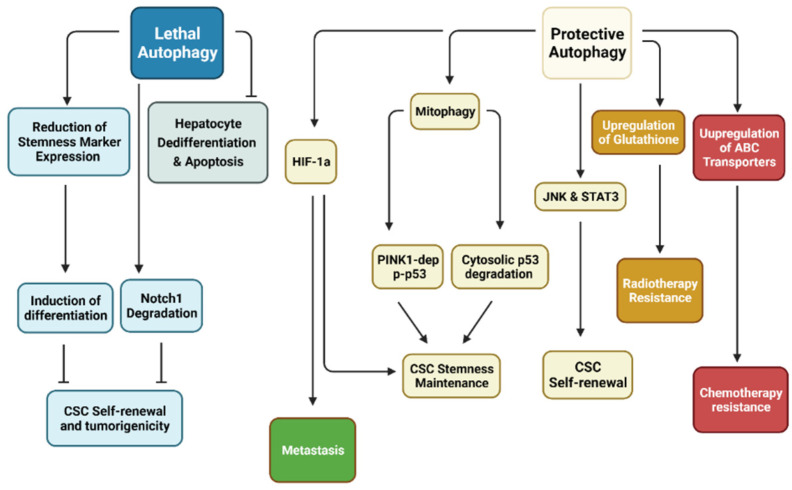
Relationship between autophagy with CSCs. Research has demonstrated that autophagy may be separated into two distinct categories in cancer stem cells (CSCs). Protective autophagy enhances CSC characteristics, including stemness maintenance through Pten-induced putative kinase 1 (PINK1)-dependent inactivation of p53 but also degradation of cytosolic p53. CSC self-renewal is dependent on Jun N-terminal kinase (JNK) and signal transducer and activator of transcription 3 (STAT3). Hypoxia-inducible factor-1α (HIF-1α)-dependent autophagy has been shown to enhance CSC stemness while promoting metastasis. CSC resistance to various treatment modalities revolves around upregulation of antioxidants to remove reactive oxygen species (ROS) and ATP-binding cassette (ABC) transporters to efflux anti-cancer drugs. On the other hand, lethal autophagy inhibits CSC self-renewal through degradation of Notch1 and reduction of stemness markers. Inhibition of this form of autophagy has been shown to induce tumorigenesis. (Created with BioRender.com, accessed on 11 October 2021).

**Table 1 cancers-14-00381-t001:** Active clinical trials targeting cancer through regulation of autophagy.

Identifier	Cancer Type	Intervention	Phase	CSC Specific?
NCT03979651	Melanoma	Chloroquine & kinase inhibitor	NA	NO
NCT03513211	Prostate Cancer	Hydroxychloroquine & antifungal drug	II	NO
NCT03037437	HCC	Hydroxychloroquine & kinase inhibitor	II	NO
NCT04841148	Breast Cancer	Hydroxychloroquine & kinase inhibitor	II	NO
NCT04735068	NSCLC	Hydroxychloroquine & kinase inhibitor	II	NO
NCT04132505	PAAD	Hydroxychloroquine & kinase inhibitor	I	NO
NCT04316169	Breast Cancer	Hydroxychloroquine & kinase inhibitor	I	NO
NCT04214418	GIC	Hydroxychloroquine & kinase inhibitor & ICB	I/II	NO
NCT04524702	PAAD	Hydroxychloroquine & Vitamin D analog	II	NO
NCT04341207	Cancer & COVID-19	Hydroxychloroquine & antibiotic	II	NO
NCT03774472	Breast Cancer	Hydroxychloroquine & kinase inhibitor & aromatase inhibitor	I/II	NO
NCT03825289	Pancreatic Cancer	Hydroxychloroquine & kinase inhibitor	I	NO
NCT04145297	GIC	Hydroxychloroquine & kinase inhibitor	I	NO
NCT04566133	Biliary Cancer	Hydroxychloroquine & kinase inhibitor	II	NO
NCT03377179	CCA	Hydroxychloroquine & kinase inhibitor	II	NO
NCT04593758	CCS	Hydroxychloroquine & anti-mitochondrial drug	I/II	NO
NCT04911816	PAAD	Hydroxychloroquine & FOLFIRINOX	I/II	NO
NCT03598595	Osteosarcoma	Hydroxychloroquine & taxane & nucleoside	I/II	NO
NCT04201457	Glioma	Hydroxychloroquine & enzyme/kinase inhibitors	I/II	NO
NCT03979651	Melanoma	Hydroxychloroquine & kinase inhibitor	NA	NO
NCT03008148	Glioblastoma	Hydroxychloroquine & alkylating agent & radiotherapy	II/III	NO
NCT04375813	Bladder Cancer	Rapamycin	II	NO
NCT03439462	CRC	Rapamycin	I/II	NO
NCT02389309	Brain Cancer	Rapamycin & alkylating agent & kinase inhibitor	I	NO
NCT03662412	Pancreatic Cancer	Rapamycin	I/II	NO
NCT03433183	MPNST, NF	Rapamycin & kinase inhibitor	II	NO
NCT03571438	Kidney Cancer	Rapamycin & kinase inhibitor	NA	NO
NCT00700258	RCC, MCL, GIC	Rapamycin & kinase inhibitor	NA	NO
NCT02642094	Breast cancer	Rapamycin	II	YES

Abbreviations: Hepatocellular Cancer (HCC), Non-Small Cell Lung Cancer (NSCLC), Pancreatic Adenocarcinoma (PAAD), Gastrointestinal Cancer (GIC), Immune checkpoint blockade (ICB), Cholangiocarcinoma (CCA), Clear Cell Sarcoma (CCS), Colorectal Cancer (CRC), Malignant Peripheral Nerve Sheath Tumors (MPNST), Neurofibromatosis (NF), Renal cell carcinoma (RCC), and Mantle-Cell Lymphoma (MCL).

**Table 2 cancers-14-00381-t002:** Preclinical data of the effects of pharmaceutical and genetic inhibition of autophagy on cancer stemness.

Author	Mechanism of Action	Cell Line	Animal	Results
Sharif et al., (2017) [62]	NAMPT inhibition	Teratocarcinoma CSCs	NO	↓ POU5F1, Nanog, & SOX2 expression
Sharif et al., (2017) [62]	ATG12 KD & ATG7 KD	HNSCC CSCs	NO	↓ Stemness ↑ Differentiation
Pagotto et al., (2017) [2]	ATG5 KO, CQ	Ovarian CSCs	NO	↓ Spheroid formation ↓ Stemness markers
Li et al., (2017) [64]	ATG3 KD, ATG7 KD & CQ	Axin2+CD90+ CSCs	NO	↓ HGF expression ↓ Stemness markers
Wang et al., (2021) [61]	3-MA, BafA1 & Rapamycin	Lung CSC A549	NO	↑ CSC stemness ↑ Spheroid/tumor formation
Vazquez-Martinet al., (2016) [69]	PINK1 KD	iPSC	NO	↓ Mitochondrial rejuvenation ↑ Differentiation
Liu et al., (2017) [71]	ATG5 KD, ATG7 KD, 3-MA	Hepatic CSCs	NO	↑ Phosphorylated p53 ↓ Nanog expression
Zhu et al., (2014) [5]	HIF-1A siRNA, 3-MA	Pancreatic CSCs	NO	↓ Vimentin & MMP-9 ↑ Epithelial phenotype ↑ E-cadherin expression
Digomann et al., (2019) [72]	ATG5 KD & BafA1	HNSCC CSCs	NO	↑ Radiosensitivity ↑ Apoptosis
Yang et al., (2021) [73]	Irradiation	Breast CSCs	NO	↑ Autophagic vesicles ↓ Initial apoptosis induction
Zhu et al., (2021) [6]	Irinotecan (chemotherapy) w/ CQ	NO	Mouse	↓ Tumor size
Ma et al., (2021) [74]	Taxol w/ Beclin1/ATG5 KD	Radio-resistant Bladder CSCs	NO	↑ Apoptosis
Brunel et al., (2021) [4]	Temozolomide (chemotherapy) w/ Beclin1 KD	GSCs	NO	↓ Proliferation No change in apoptosis
Zhu et al., (2021) [6]	Rapamycin	CSCs	NO	↑ Chemoresistance & stemness Sox2 upregulates ABC transporters
Zhu et al., (2021) [6]	2-MA	CSCs	NO	↓ Malignant cancer phenotype
Liu et al., (2021) [75]	Afatinib (RTK inhibitor) w/ 3-MA or CQ	HNSCC CSCs	NO	↑ Afatinib-induced apoptosis with coincubation
Zhuang et al., (2012) [76]	Curcumin	GSCs	NO	↓ Self-renewal ↑ Induction of differentiation
Tao et al., (2018) [50]	AZD8055 or rapamycin	GSCs	NO	↓ Self-renewal ↓Tumorigenicity
Tao et al., (2018) [50]	mTOR inhibition	NO	Mouse	↓ Tumor size and prolonged survival
Tao et al., (2018) [50]	Beclin KD, ATG5 KD & CQ	GSCs	NO	↑ Stemness markers ↑ Proliferation & clonogenicity
Barthet et al., (2021) [51]	ATG5KD /ATG7 KD	LPCs	Mouse	↑ TAZ & YAP co-expression ↑ Ductular cell formation ↑ Carcinogenesis

Abbreviation: Increased (↑), decreased (↓) knockdown (KD), head and neck squamous cell carcinoma (HNSCC), glioblastoma stem cell (GSC), liver progenitor cells (LPCs), chloroquine (CQ), receptor tyrosine kinase (RTK).

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
