# Peer review of "Autophagy Regulation on Cancer Stem Cell Maintenance, Metastasis, and Therapy Resistance"

_cancers, 2022, doi:10.3390/cancers14020381_

Round 1

Reviewer 1 Report

Wang et al. present a review summarizing the roles of autophagy in cancer stem cells. It is a balanced review with an emphasis on pro-tumorigenic roles of autophagy but also covers autophagy mediated cell death. The review is brief and could benefit from discussion of some of the following points:

  1. Although the concept of plasticity is touched upon on line 213, it is not introduced adequately. Additionally, the co-existence of distinct CSCs within tumors should also be introduced to highlight the complexity of CSCs and plasticity in some tumor types.
  2. In section 2, the authors generally are referring to macro-autophagy. It would be beneficial to specify this and introduce other types of autophagy (i.e., micro-autophagy and chaperone-mediated autophagy).
  3. On lines 169-170, the sentence implies that the ULK1 and BECN1 complexes are complementarily redundant. However, in most cases, both are required and necessary for sequential steps in autophagy.
  4. In section 6, the authors mention seemingly discordant roles of autophagy in cancer stem cells. It would be worth elaborating with examples that such observations could be due to non-autophagy functions of some autophagy genes (e.g., LC3-associated phagocytosis, exosome secretion, etc.). Accordingly, future directions could benefit from the use of specific autophagy-deficient mutant alleles of autophagy genes, rather than knocking out the entire gene, which would also disrupt its non-autophagy functions.

Minor:

  1. There are grammatical errors that need to be corrected throughout the manuscript.

Author Response

Reviewer: 1

  1. Although the concept of plasticity is touched upon on line 213, it is not introduced adequately. Additionally, the co-existence of distinct CSCs within tumors should also be introduced to highlight the complexity of CSCs and plasticity in some tumor types.

Response: As suggested, we have added a section clarifying this point, which can be found from lines 62-73. 

2. In section 2, the authors generally are referring to macro-autophagy. It would be beneficial to specify this and introduce other types of autophagy (i.e., micro-autophagy and chaperone-mediated autophagy).

Response: Thank you for your suggestion. We have added these points, which can be found from lines 128-136.

3. On lines 169-170, the sentence implies that the ULK1 and BECN1 complexes are complementarily redundant. However, in most cases, both are required and necessary for sequential steps in autophagy.

Response:Thank you for the clarification. These changes can be found from lines 145-148.

4. In section 6, the authors mention seemingly discordant roles of autophagy in cancer stem cells. It would be worth elaborating with examples that such observations could be due to non-autophagy functions of some autophagy genes (e.g., LC3-associated phagocytosis, exosome secretion, etc.). Accordingly, future directions could benefit from the use of specific autophagy-deficient mutant alleles of autophagy genes, rather than knocking out the entire gene, which would also disrupt its non-autophagy functions.

Response: Thank you for the suggestion. This is an interesting point, and the alterations can be found on lines 369-380.

Reviewer 2 Report

The manuscript “Autophagy Regulation on Cancer Stem Cell Maintenance and Function” describes bidirectional role of autophagy in cancer stem cell. Showing multiple scientific studies, the authors suggest that unraveling the regulation of autophagy may offer novel treatment to effectively eliminating CSCs and minimize cancer relapse. Overall, this review discusses a highly interesting topic to the scientific cancer community, but several things need to be supplemented to be more well-organized review.

  1. The word ‘Function’ of the title is too comprehensive compared to the main text. I understood you only focused on the self-renewal, maintenance of a stemness state, and treatment resistance of cancer stem cells so how about changing ‘function’ in a different word?
  2. The contents of lines 169 to 172 are unnecessary because they have already been well explained in the previous paragraphs. If you want to keep that paragraph, it would be better to combine the paragraph with previous paragraphs in section 2.
  3. From line 248 to line 254, there are 3 explanations about the combination of chemotherapy and autophagy inhibition. It would be better to unify the structure of these phrases; Chemotherapy (Inrinotecan) / Taxol (chemotherapy) / Temozolomide chemotherapy treatment. Additionally, please give brief type or mechanism of action about the preceding chemotherapeutic agents.
  4. In conclusion, the authors hypothesized that the type of autophagy differs based on the cell type, tumor stage, TME, and interaction with the apoptosis pathway (Line 317). However, there is one explanation for the relationship between CSCs and TME in this review article. Please discuss the mechanism of how TME affects autophagy in CSCs.
  5. Please provide more explanations about why CSCs resist conventional cancer therapies. 
  6. Line 178 - 185 - there is no reference. 
  7. Line 188 - the reference is incorrect. Please check the reference number throughout the manuscript. 
  8. There is a typo in Figure 2. 

Author Response

1. The word ‘Function’ of the title is too comprehensive compared to the main text. I understood you only focused on the self-renewal, maintenance of a stemness state, and treatment resistance of cancer stem cells so how about changing ‘function’ in a different word?

Response: We have changed the title to be more specific.

2. The contents of lines 169 to 172 are unnecessary because they have already been well explained in the previous paragraphs. If you want to keep that paragraph, it would be better to combine the paragraph with previous paragraphs in section 2.

Response: We apologize for the confusion. There was some mix up when the figure was moved around. Lines 169-172 are the legend to Figure 1. We have fixed this in the updated version.

3. From line 248 to line 254, there are 3 explanations about the combination of chemotherapy and autophagy inhibition. It would be better to unify the structure of these phrases; Chemotherapy (Inrinotecan) / Taxol (chemotherapy) / Temozolomide chemotherapy treatment. Additionally, please give brief type or mechanism of action about the preceding chemotherapeutic agents.

Response: Thank you for this suggestion. We have added these changes, which can be found in lines 278-280, 282-284, 288-292.

4. In conclusion, the authors hypothesized that the type of autophagy differs based on the cell type, tumor stage, TME, and interaction with the apoptosis pathway (Line 317). However, there is one explanation for the relationship between CSCs and TME in this review article. Please discuss the mechanism of how TME affects autophagy in CSCs.

Response: Thank you for bringing this to our attention. We have added a section that talks about how hypoxia in the TME induces autophagy and how that affects CSC characteristics. This section can be found from lines 244-257.

5. Please provide more explanations about why CSCs resist conventional cancer therapies. 

Response: The paper discusses 2 methods by which CSCs are able to utilize autophagy to resist conventional cancer therapies. First is the upregulation of glutathione, an antibody to prevent genomic oxidative damage due to radiotherapy. Second, irradiation-induced autophagy activation delays the onset of apoptosis activation; however, the mechanism of action is unknown and need further clarification. Third, cotreatment of autophagy inhibition, whether via genetic or pharmaceutical inhibition, with various types of chemotherapy was the most effective at reducing CSC viability/growth. A possible mechanism of action would be autophagy-dependent upregulation of ABC transporters that work to efflux chemotherapy drugs out of the cell before the drugs are able to take effect.

6. Line 178 - 185 - there is no reference. 

Response: We apologize for the misunderstanding. The whole paragraph is referring to the same study.

7. Line 188 - the reference is incorrect. Please check the reference number throughout the manuscript. 

Response: We apologize for the confusion. We have checked the reference numbers throughout the manuscript and fixed the error (line 201)

8. There is a typo in Figure 2. 

Response: Thank you for pointing this out.

Reviewer 3 Report

Review of the manuscript entitled

‘Autophagy Regulation on Cancer Stem Cell Maintenance and Function’

by Xin Wang, Jihye Lee and Changqing Xie

The authors in the manuscript entitled ‘Autophagy Regulation on Cancer Stem Cell Maintenance and Function’ review the role of autophagy in cancer stem cells (CSCs) in the cancerogenesis and its implications for clinical treatments. The manuscript is well written, however, it requires minor language corrections. In the reviewers’ opinion the manuscript could be improved according the issues enlisted below:

  1. Fig. 1 depicts modification of LC3 during the formation of autophagosome. Please pay attention that it can be concluded that LC3 generates LC3-I and PE from the Fig. 1.
  2. Please consider expanding the figure legends to the common standard where figures are treated as separate items from the manuscript. Thus figure legends should include detailed description explaining all abbreviations depicted in the figure.
  3. The manuscript covers the presented problem in a too shallow way and thus it loses the critical approach. The reviewer suggests to develop the manuscript in the following directions, especially that the authors mention these issues throughout manuscript:
  • the role of autophagy in CSCs’ migration and invasion,
  • the role of autophagy in CSC’s chemoresistance and immunoresistance,
  • the role of mitophagy and the link between mitophagy, Warburg effect and oxidative phosphorylation in CSCs,
  • the role of microenvironmental autophagy for maintenance of CSCs niche,
  • the future perspectives presenting difficulties in the introduction of autophagy inhibitors into clinical treatment as it could be a double-edged sword.

4. Data presented in Tab. 1 demonstrate that clinical trials are not directed toward targeting CSCs through autophagy modulation. It would be advisable to include Table presenting preclinical in vitro studies showing results of co-treatment including autophagy inhibitor and anticancer treatment.

Author Response

  1. Fig. 1 depicts modification of LC3 during the formation of autophagosome. Please pay attention that it can be concluded that LC3 generates LC3-I and PE from the Fig. 1.

Response: Thank you for bringing this to our attention. The new Fig.1 shows PE being added to LC3-I with an arrow.

2. Please consider expanding the figure legends to the common standard where figures are treated as separate items from the manuscript. Thus, figure legends should include detailed description explaining all abbreviations depicted in the figure.

Response: We have made the suggested changes which can be found from lines 157-165.

3. The manuscript covers the presented problem in a too shallow way and thus it loses the critical approach. The reviewer suggests to develop the manuscript in the following directions, especially that the authors mention these issues throughout manuscript:

  • the role of autophagy in CSCs’ migration and invasion,

Response: The changes can be seen from lines 266-280, which looks at how the microenvironment affects both autophagy and CSC characteristics including migration.

  • the role of autophagy in CSC’s chemoresistance and immunoresistance,

Response: I believe the paper does include chemoresistance which can be found form lines 301-328. 

  • the role of mitophagy and the link between mitophagy, Warburg effect and oxidative phosphorylation in CSCs,

Response: Thank you for the suggestion. However, these topics were only mentioned as a brief introduction to CSCs and are not the focus of discussion. They are outside of the scope of this literature review.

  • the role of microenvironmental autophagy for maintenance of CSCs niche,

Response: The changes can be seen from lines 266-280, which looks at how the microenvironment affects both autophagy and CSC characteristics including migration.

  • the future perspectives presenting difficulties in the introduction of autophagy inhibitors into clinical treatment as it could be a double-edged sword.

Response: Thank you for the suggestion. We added a new section that discusses autophagy genes that also have non-autophagy related functions. This section can be found from lines 404-417.

4. Data presented in Tab. 1 demonstrate that clinical trials are not directed toward targeting CSCs through autophagy modulation. It would be advisable to include Table presenting preclinical in vitro studies showing results of co-treatment including autophagy inhibitor and anticancer treatment.

Response: Thank you for the suggestion. We have added a table that summarizes the findings of each study that was covered in the review. There are only a few studies that cotreat with an autophagy inhibitor and anticancer treatment. This table can be found from lines 260-263.

Reviewer 4 Report

The review is comprehensive but can benefit further from the below minor suggestions:

1) Each sections which does not have accompanying table summarizing seminal studies in that section should have such a summary for readers.

2) The authors might want to include a section highlighting studies showing tumor autophagy's connection to the immune cells of the TME.

3) Therapeutic or genetic blockade of pathways involved in autophagy and how it affects tumorigenesis should be summarized into a table.

Author Response

1. Each sections which does not have accompanying table summarizing seminal studies in that section should have such a summary for readers.

Response: Thank you for the suggestion. We have added a table that summarizes the findings of each study that was covered in the review and can be found from lines 260-263.

2. The authors might want to include a section highlighting studies showing tumor autophagy's connection to the immune cells of the TME.

Response: This topic was introduced at the future directions as another path of investigation. We have removed this line as the topic is outside of the scope of this literature review.

3. Therapeutic or genetic blockade of pathways involved in autophagy and how it affects tumorigenesis should be summarized into a table.

Response: Thank you for the suggestion. We have added a table that summarizes the findings of each study that was covered in the review and can be found from lines 260-263.